# Shore Evidences of a High Antarctic Ocean Wave Event: Geomorphology, Event Reconstruction and Coast Dynamics through a Remote Sensing Approach

Stefano Ponti and Mauro Guglielmin *

Department of Theoretical and Applied Sciences, Insubria University, 21100 Varese, Italy; s.ponti@uninsubria.it
* Correspondence: mauro.guglielmin@uninsubria.it

**Abstract:** Remote sensing can be helpful in defining the dynamic of a high-latitude coastal environment where the role of cryogenic processes like sea-ice or permafrost are the main drivers together with storm surge and wind action. Here we examined the geomorphological dynamics of a beach located at Edmonson Point (74° S) not far from the Italian Antarctic Station "Mario Zucchelli" between 1993 and 2019 using different remote sensing techniques and field measurements. Our data demonstrate that the average rate of surficial increase of the beach ($0.002 \pm 0.032$ m yr$^{-1}$) was slightly higher than the uplift rate determined by previous authors (0–1 cm yr$^{-1}$) in case of pure isostatic rebound. However, we suggest that the evolution of EPNB is likely due to the couple effect of vertical uplift and high wave-energy events. Indeed, the coastline accumulation could be related to the subsurface sea water infiltration and annually freezing at the permafrost table interface as aggradational ice as suggested by the ERT carried out in 1996. This ERT suggests the occurrence of saline frozen permafrost or hypersaline brines under the sea level while permafrost with ice occurred above the sea level. The beach also revealed areas that had quite high subsidence values (between 0.08 and 0.011 m yr$^{-1}$) located in the area where ice content was higher in 1996 and where the active layer thickening and wind erosion could explain the measured erosion rates. Here, we also dated at the late morning of 15 February 2019 coastal flooding and defined a significant wave height of 1.95 m. During the high oceanic wave event the sea level increased advancing shoreward up to 360 m, three times higher than the previous reported storm surge (81 m) and with a sea level rise almost five times higher than has been previously recorded in the Ross Sea.

**Keywords:** beach processes; coastal storm; coastal geomorphology; sea ice; Antarctica

## 1. Introduction

Coastal changes have a great impact on human development [1]. Coasts are generally studied for their erosion [2,3], retreat [4,5], or as ecosystem services [6]. Coastal changes at mid latitudes principally happen when high tide couples with high wave activity [7]. High-latitude (above the Arctic and Antarctic circle) coastal geomorphology additionally highlights the role of cryogenic phenomena like sea-ice [8,9] or permafrost [10–13] as the major components of shore dynamics. Here, the cryogenic nature of shorelines could trigger impacts on global carbon release [14] and local communities [15]. Sea-ice can have a protective, constructive, or erosional effect on polar beaches depending on the bathymetry and underlying topography [16]. For example, the sea-ice cover decreases the wave effect onshore [17–19]. On the contrary, ice-rich permafrost coasts have been found to increase the retreating rate [11,20,21]. Despite the long ice-cover period in polar regions, high coastal activity is visible when sufficient open-water fetch is available [16] and changes can occur in short periods [22] when storm surges occur [11]. Indeed, sea surges are more likely to impact lengthened open-water seasons due to the increased fetch [23]. Consequently, studies are focused mainly on the reconstruction of storm events from the beach's geomorphology [16,24]. Storm surges in polar regions have been found to

accelerate coastal retreat up to 19 m per single event [11], which may also underestimate the recession rate all over the arctic coasts [25]. Recently, the availability of ocean dynamic models such as HYCOM (https://www.hycom.org/) and WAVEWATCH III® (https://polar.ncep.noaa.gov/waves/wavewatch/) provide temporally and spatially continuous global data for the reconstruction of storm surges.

However, there is a lack of knowledge of sea-ice and coastal feature relationships, especially when both move and develop after a short storm event. In Antarctica, only Dayton et al. [26] described the effects of a flood event in Southern Victoria Land on the coast, while Simeoni et al. [27] qualitatively described some beaches of the Northern Victoria Land including our study area. Other studies have focused on sea-ice—such as coast interactions, e.g., [16,19,28]—but few of them have focused on Antarctic shorelines [24,29]. Here, the beach dynamics are related not only to cryogenic processes but also to the coupled effect of the wave energy and the vertical uplift of the land [30].

Isostatic rebound of Antarctic coasts is well-known, e.g., [31–33] and one of the most evident traces of isostatic rebound on beaches are raised beaches [34]. These are well studied in Victoria Land [35–37] and definitely one of the slow-rate processes that changes Antarctic coast morphology [29].

Remote sensing provides a solution not only for coastal geomorphological mapping [38] and temporal changes [39] in harsh environments like Antarctica, but also for classifying the fast ice [40] or land cover features [41]. Among all remote sensing techniques, Structure from Motion (SfM) on coasts has been used for mapping moss beds [42] or sea-ice [43], but not for the dynamic or morphology of Antarctic beaches. This paper aims to: (1) analyze the dynamic of the beach since 1993; and (2) describe the effects of a single high oceanic wave event that occurred during the austral summer of 2019 and was detected through remote sensing in the Antarctic coastal environment Edmonson Point North Beach (EPNB, Victoria Land, Antarctica).

## 2. Materials and Methods

### 2.1. Study Area

Edmonson Point (74°20′ S, 165°08′ E) is a volcanic ice-free area located in Wood Bay on the west coast of the Ross Sea, northern Victoria Land, Continental Antarctica. The landscape is dominated by Mt. Melbourne, a dormant volcano. The whole area is lower than 300 m a.s.l. with a maximum tidal amplitude of 19.6 cm [44] and presents evidences of glacial [45] and periglacial [46] activity (Figure 1). The volcanic activity of Mt. Melbourne produced a dark substrate composed of basaltic lavas, scoria, pumice, and tuff related to the McMurdo Volcanic Group, generated from the Cenozoic eruptive activity of Mt. Melbourne [47]. Part of the Mt. Melbourne glacier descending towards the sea in the area of Edmonson Point has built a complex of ice-cored moraines along the borders of the glacier. These moraines are composed of coarse angular volcanic clasts and, locally, of well-rounded pebbles in a matrix of volcanic sand [45].

The area is characterized by alluvial sediments and weathered basaltic outcrops with several raised beaches, locally pitted, and the highest berm is 6 m a.s.l. [45]. The flat beach area is generally composed of pebbles and ashes: the innermost beach is composed of medium-fine sand, while the short and steep slope against the sea of pebbles does not contain sand. Boulders can be found on the structural reliefs and decrease in size seaward and shoreward. The more internal depressed area is delimited by a relict shoreline and terminates shoreward with a big pond [27]. Nivo-aeolian deposits and ventifacts have been individuated beyond the frost fissuring of finer deposits [46].

With respect to the rest of the coastal areas of Northern Victoria Land, the area of Edmonson Point is well sheltered from local katabatic winds at the innermost areas. There, the summer winds principally come from the East with daily maxima up to 6–10 knots, occasionally 25–35 knots [48]. Monthly mean air temperature ranges between −25.9 °C (August) and −0.1 °C (January). Despite the general aridity, it is possible to encounter several meltwater streams, ponds, and lakes that make the Edmonson Point area one of the

largest freshwater networks of Victoria Land. An overview of the climatic data available in
this area can be found in Scarchilli et al. [49]. Where sea level records have been recorded
but not published; more information is available at https://www.geoscience.scar.org/
geodesy/perm_ob/tide/terranova.htm.

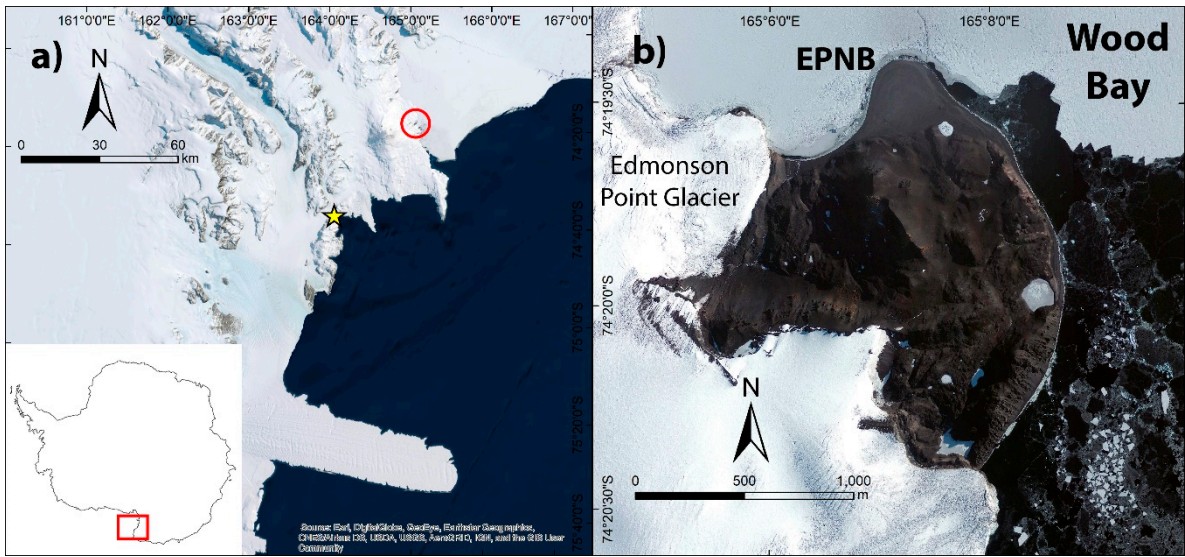

**Figure 1.** Study area: (**a**) Victoria Land (red square) and Edmonson Point location (red circle) next to Mt. Melbourne,
(**b**) Edmonson point area and Edmonson Point North Beach (EPNB) position. Yellow star indicates Maria AWS.

In this climate, permafrost is continuous everywhere and the active layer thickness
ranges between 23 to 55 cm [50,51]. The vegetation of Victoria Land is composed exclusively
of cryptogams and a detailed floristic list of the study area has been made by Cannone [52].

Due to the ecological importance of this site, Edmonson Point was designated an
Antarctic Specially Protected Area (ASPA) No. 165 in June 2006.

### 2.2. Field Data and Morphometry

The field work consisted of measurements of the mapped geomorphic elements such
as the height of sea-ice blocks (SIBs) and the depth of holes formed by sea-ice blocks,
in both cases taken with a ruler (1 cm of resolution, 5 cm of accuracy). Estimation of
the aeolian action was obtained measuring the aeolian sediment accumulated within the
manipulation experiments (that acted as a wind barrier) with a ruler (1 cm of accuracy)
during the Antarctic summer of 2018 and was related to the entire previous year.

Temperature loggers (Hobo U23-003, accuracy of 0.21 °C) were set on the beach during
previous manipulation experiments in order to record the difference of surface (2 cm depth)
soil temperature of the International Tundra Experiment (ITEX) that consisted in open-top
chambers (OTCs), a precipitation shield (TT), and a snow fence (SF) [53]. Two thermistors,
placed horizontal to the surface in different manipulation sites (OTC and TT), were chosen
to underline the beginning of the storm event and the inundation ending. The thermistor
in the OTC (5.4 m of elevation) was protected from external agents by the chamber shield,
while the one in the TT (3.2 m of elevation) was not protected at the sides.

The wind data were obtained from the automatic weather station (AWS) "Maria"
(mount Browning, 74° 37′ 35″ S, 164° 0′ 40″ E, 355 m a.s.l.), which is the most representative
of the sea sector in front of the beach. Data and information were obtained from 'Meteo-
Climatological Observatory at Mario Zucchelli Station (MZS) and Victoria Land' of the
National Antarctic Program (PNRA)-http://www.climantartide.it.

The features' morphometry calculated from the orthophoto and DEM consisted of
each feature's: height-depth (m) as difference from top and bottom elevation, length (m)
and width (m) as planar distances, area (m²), longest axis orientation (°), elevation of the

feature's location (m), distance from the coastline (m), and z-range sea surface (ZSS) (m). The latter represents the maximum elevation of the significant wave that transported the SIB; it has been computed through two different calculations: (1) if the SIBs or the landforms related to them are placed on the seaside of the main ridge, ZSS is the difference of elevation between the features and the sea ice level; (2) if the SIBs or related landforms are on the shoreward side of the main ridge, the ZSS is the difference between the elevation of the main ridge and the features' elevation measured along the direction from the feature to the coastline. All the elevations in this study refer to altitudes above the lowest depression of the DEM (on the sea ice) that has been set to zero. Moreover, slope and profile curvature calculated in ArcGIS helped in delineating the beach ridges [45]. A qualitative grain size analysis was also visually conducted to understand the spatial distribution of the clast dimensions.

### 2.3. Aerial Survey

On 9 December 2019 a helicopter survey was conducted above the EPNB similarly to Dietrich [54], except for the fact that the camera was manually held outside the window. The survey consisted of a 10-minute-flight at 100 m of altitude above ground level that manually simulated a pre-planned photo mosaic drone flight. A smartphone camera was used during the flight time and recorded a 4K video at quasi-nadir position. The nadir frames ($3840 \times 2160$ pixels of resolution) were extracted from the video at 2 s intervals to reach a total of 635 images. On average, the frames overlapped 80% (y) and 50% (x).

The images have been processed in Agisoft Metashape Professional (version 1.5.2) using a standard workflow, e.g., [55] that generated an orthophoto and DEM with a ground sample distance (GSD) of 2.5 and 5.0 cm px$^{-1}$ for the orthophoto and DEM, respectively.

Beyond the helicopter survey, 3 sequenced aerial images (available at https://earthexplorer. usgs.gov/) dated 23 November 1993 were processed in the same software. Despite having fewer images and a lack of optimal workflow characteristics [56], an overlap of 35% was guaranteed and thus the same workflow, e.g., [55] was applied here. The bundle adjustment produced an orthophoto and DEM with a GSD of 12.3 and 24.5 cm px$^{-1}$, respectively.

Six markers were selected as easily identifiable spots on the field and in order to geo-reference the helicopter point cloud, a portable Global Positioning System (GPS) Magellan ProMark 3 with a sub-metric absolute accuracy was used. Only 4 markers with the lowest reprojection error were used as control points, while the other 2 were used as check points.

The point cloud from 1993 was registered on the helicopter cloud through easily identified spots such as big stable blocks or unchanged rock outcrops. Subsequently, a cloud-to-cloud distance process (M3C2 plugin) [57] was conducted in CloudCompare 2.11 in order to spatially distribute the uncertainties, significant changes, and M3C2 distances more accurately than a DEM of Difference (DoD) [58]. Afterwards, the resolution of the 2 DEMs were down-scaled to 0.5 m in order to get less artifacts and higher accuracy [59,60].

The root mean square error (RMSE) of the check points was not useful to understand the real model error due to the low GPS accuracy. However, the relative (or geomorphological) accuracy is more important than the absolute accuracy to assess the model quality [59].

The most important clouds' and GCPs' parameters have been summarized in Table 1.

**Table 1.** Point cloud characteristics of the two aerial surveys.

| Point Cloud | Images | Dense Cloud Points | Tie Point Reprojection Error Range (px) | Markers (of Which Control Points) | Markers Reprojection Error (px) | RMS Reprojection Error (cm) | Orthophoto GSD (cm/px) | DEM GSD (cm/px) |
|---|---|---|---|---|---|---|---|---|
| Helicopter (9 December 2019) | 635 | 132.5 M | 1.2–5.5 | 6 (4) | <0.05 | 19.6 | 2.5 | 5.0 |
| Past (23 November 1993) | 3 | 23.4 M | 0.6–0.7 | 7 (7) | 0.6 | 14.9 | 12.3 | 24.5 |

Measurements of the absolute accuracy were taken at the manipulation experiments for the 3D model calibration: (a) known distances of the manipulation experiments were matched with distances from the orthophoto to assess a planar accuracy of the model, while (b) the vertical accuracy was obtained through the length of shadows. Knowing the height of one operator in the orthophoto (1.80 m) and its shadow length (2.10 m), it has been possible to randomly select SIBs placed on flat areas to proportionally calculate their heights from their shadow lengths. The RMSE of the differences between measured and modelled planar/vertical distances was used as the quality assessment of the model [61]. Concerning the propagation of errors during the cloud-to-cloud distance process, a limit of detection (LoD) was considered for the final error at a 95% confidence interval according to James et al. [62].

### 2.4. Satellite Data and High Oceanic Wave Event Reconstruction

Seven Landsat panchromatic images (ranging from 3 February 2019 to 28 February 2019) downloaded from https://earthexplorer.usgs.gov/ (30 m of resolution) were utilized to identify: (a) the sea ice breakup and the consequent initiation of the wind effect on the open sea surface, and (b) the moment the coast was overwashed [63] during which the ice blocks invaded the coast. Landsat images were also utilized to calculate the fetch as the length from the coast to the closest huge mass of consistent sea ice. The fetch was then used to calculate the significant wave height, directly proportional to the beach total swash (wave run-up) [64] and thus indicating the increased water level [65] through the JONSWAP experiment (Equation (1)) [66]:

$$Hs = 0.0016g^{-0.5} X^{0.5} U \tag{1}$$

where Hs is the significant wave height (m), g is the gravitational acceleration (m s$^{-2}$), X is the fetch length (m) and U is the wind speed (m s$^{-1}$).

The validation of Hs was conducted comparing both the sea surface height (SSH) derived from the HYCOM model (available at https://www.hycom.org/) and the Hs derived from the WAVEWATCH III® system (available at https://polar.ncep.noaa.gov/waves/wavewatch/) at the same time of the high oceanic wave event occurrence. Both systems include data from global circulation models and buoys; therefore, they have a complete climatic assessment of the levels of the studied event. The comparison consisted of calculating zonal statistics of a polygon (260 × 116 km) that resembled the models' pixels (for HYCOM and for WAVEWATCH III®) closest to the study area and not farther than 190 km.

## 3. Results

### 3.1. Geomorphology

The wave run-up shoreward limit was reconstructed by passing next to the upper limit of the mapped landforms or at damaged experiment sites (SF, OTC2, and TT).

The principal geomorphological structure was the main raised beach ridge which is clearly evident, continuous, and located at an elevation of 7.5 ± 1.2 m. However, the 5 innermost (older) beach ridges were less pronounced (Figure 2), not continuous and located at the respective elevations of 7.7 ± 0.6, 7.5 ± 0.4, 6.5 ± 0.4, 6.0 ± 0.3, 4.5 ± 0.1 m. Among the smaller landforms that occur on the beach, some are erosional like erosion scarps and rill troughs that were generated by the sea water inundating the beach, while the kettle holes (KHs) are related to the weight of both the melted out or still present SIBs. SIBs were mapped separately when stand-alone or associated with the KHs, similar to the KHs.

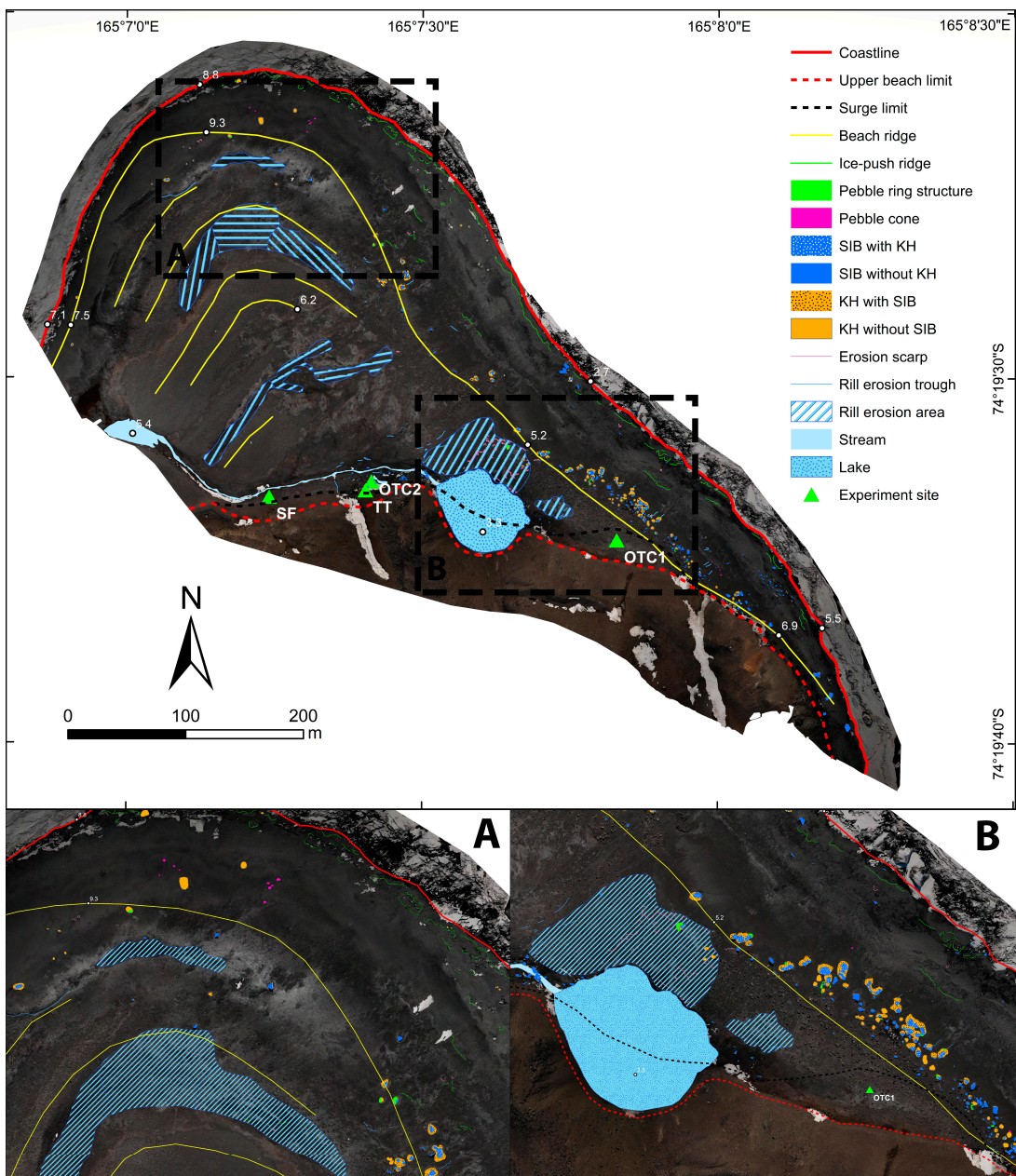

**Figure 2.** Geomorphological map of EPNB drawn upon the orthophoto obtained through the helicopter images. SIB = sea-ice block, KH = kettle hole. SIB with KH and KH with SIB are kept separately in order to differentiate their morphometry in the next section. The experiment sites show the positions of 2 open top chambers (OTC1-2), the precipitation shield (TT) and the snow fence (SF). Please note that "Coastline" refers to maximum extension of sea-ice that might not coincide with the real coastline. The rill erosion area symbol (oblique lines) matches the real orientation of the landforms.

Among the aggradational landforms, the following were recognized: ice-push ridges (directly affected by the sea-ice pressure), pebble cones, and pebble ring structures. The latter are small landforms (2.4 m of mean length) that resemble the features found by Urdea [67] on the coast of the Svalbard Islands, referred to as frost mounds phenomena. The distribution of the smaller landforms was not random but more concentrated around the lake or at the seaward side of the southern part of the main raised beach ridge where the beach is less extended. While SIBs could be found near the main ridge or around the lake, KHs could only be found along the main beach ridge (Figure 2). For other features like the ice-push ridges, their concentration, as expected, was at a maximum in the proximity of the coastline.

### 3.2. Morphometry

The morphometrical analyses are conditioned by the resulting planar RMSE (0.02 m), while the vertical RMSE was 0.1 m. As a consequence of these errors, the features with heights < 0.1 m were mapped but excluded from the calculation of their morphometrical characteristics like height or depth. Concerning the calibration of the mapped elements, the average depth of 25 depressions measured in situ ($0.31 \pm 0.1$ m) matched well the average depths of the 96 mapped depressions ($0.29 \pm 0.15$ m). Few big SIB heights were measured in situ (8) and their average height ($1.47 \pm 0.33$ m) matched quite well the average height of the biggest mapped SIBs (> median) ($1.14 \pm 0.6$ m). Aggradational forms (112) were fewer than erosional forms (163) and SIBs resulted to be the most widespread features (315 blocks).

SIBs averagely placed between 5.4–5.7 m of elevation, with accumulation forms between 5.5–8.2 m and erosional forms between 4.4–5.8 m. Areas with erosional forms averagely resulted to be larger (2.6–9.8 m$^2$) than accumulation forms (0.8–1.4 m$^2$), while SIB areas ranged between 0.8–3.2 m$^2$. Except for SIBs that had various shapes, the accumulation features were generally parallel to the coastline (ice-push ridges) or, in the case of the pebble ring structures, located north-northwestward in respect to KHs. On the contrary, erosional forms were generally orientated E–W except for the erosion scarps that had different shapes (often "V" shaped, Figure 3c). Concerning the average difference of elevation between the coastline and the forms, SIBs and KHs did not differ considerably, in fact SIBs were at 2.3 m while KHs between 1.9–2.7 m. The majority of SIBs lacked the correspondent KH (251 cases) because all the SIBs smaller than $0.28 \times 0.38 \times 0.11$ m were not able to produce enough pressure on the substrate. In addition, the KHs with SIB were larger than the only KHs.

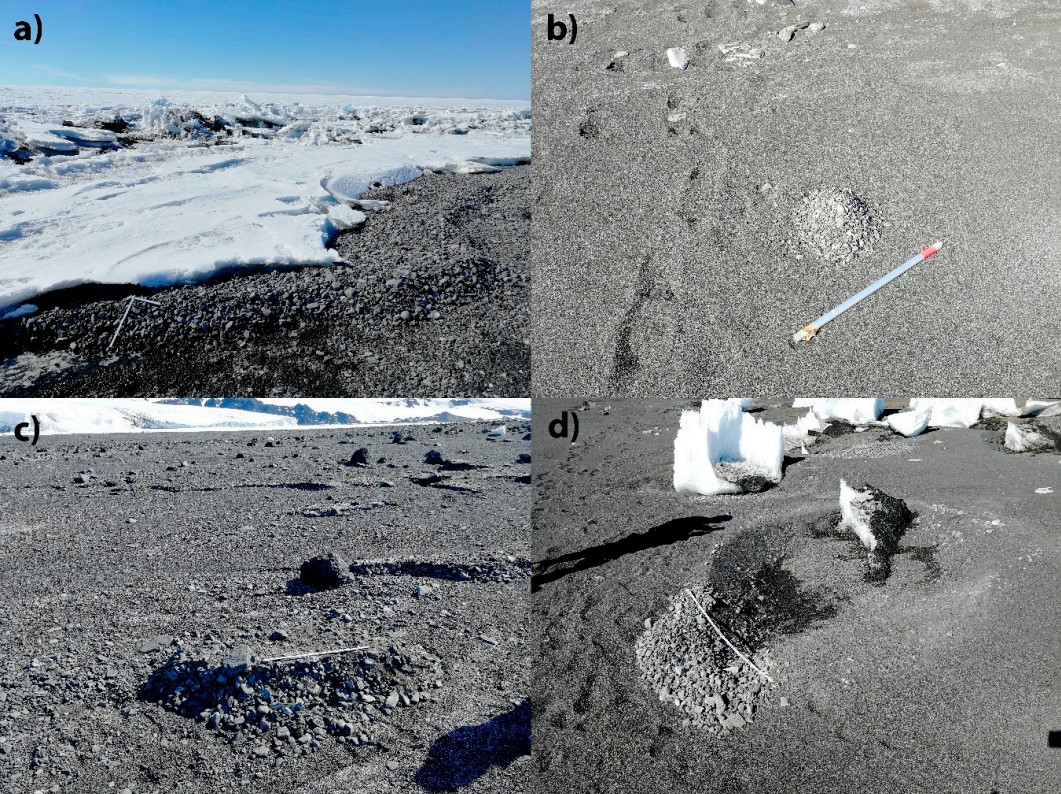

**Figure 3.** Detail of some of the mapped coastal features and the grain size classes: (**a**) ice-push ridges in class 1, (**b**) a small pebble cone in class 2, (**c**) erosion scarps in class 3, (**d**) KH with SIB and the pebble-ring structure in class 2. The ruler is extended to 80 cm in (**a**) and 100 cm in (**b**) and (**c**), while the stick in (**d**) is 70 cm long. The location of these photos is visible in Figure 4 (phots taken by S. Ponti).

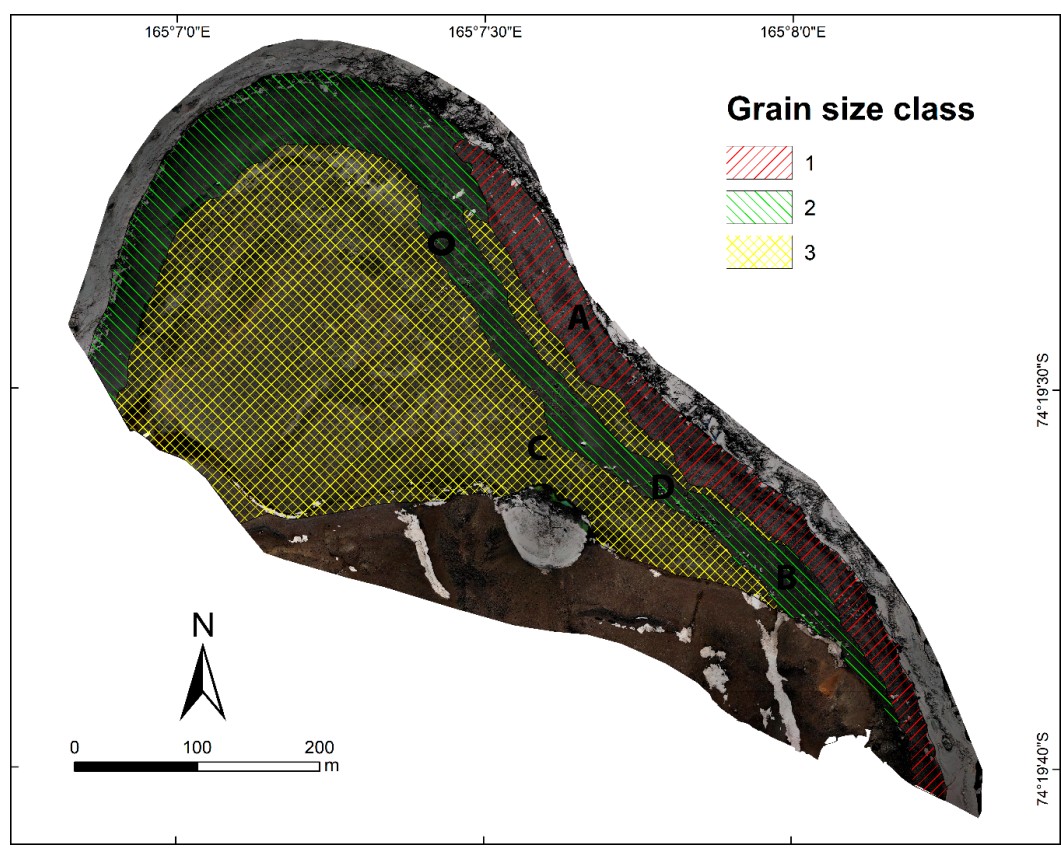

**Figure 4.** Distribution of the grain size classes at EPNB: (1) pebbles dominated, (2) sand dominated, (3) pebbles and boulders embedded in sandy matrix. Letters (A, B, C, D) refer to the location of the ground images showed in Figure 3, while the black circle refers to the location of the SIB showed in the Section 4.1.

Rill erosion areas consisted on surfaces where the density of rill erosion troughs was high and not necessarily orientated towards the maximum slope but along the flow of the incoming sea water (Table 2).

**Table 2.** Characteristics of the geomorphological features mapped on the beach. The statistics refer to medians ± standard deviations of the mapped features morphology/location. Green background color refers to aggradational features, blue to ice blocks and orange to erosion features.

| | Mapped Features | Of Which Within the RMSE | Elevation (m) | Width (m) | Length (m) | Height-Depth (m) | Distance from Coastline (m) | Area (m²) | Orientation (°)/Location | ZSS (m) |
|---|---|---|---|---|---|---|---|---|---|---|
| Ice-push ridge | 79 | 26 | 6.9 ± 2.0 | - | 6.9 ± 11.9 | 0.14 ± 0.04 | 9.1 ± 27.8 | - | Parallel to coastline | - |
| Pebble ring structure | 16 | 15 | 5.5 ± 1.4 | 0.9 ± 0.4 | 2.4 ± 1.0 | 0.28 ± 0.17 | 61.8 ± 11.9 | 1.4 ± 1.7 | from W to N of the depression | - |
| Pebble cone | 17 | 16 | 8.2 ± 1.9 | - | - | 0.23 ± 0.27 | 40.9 ± 7.6 | 0.8 ± 0.5 | - | - |
| SIB | 315 | 300 | 5.7 ± 1.6 | 1.0 ± 0.8 | 1.5 ± 1.2 | 0.5 ± 0.6 | 57.7 ± 55.1 | 1.0 ± 3.0 | 104 ± 48 | 2.3 ± 1.2 |
| SIB with KH | 64 | 64 | 5.4 ± 1.1 | 1.8 ± 0.9 | 2.7 ± 1.4 | 1.0 ± 0.5 | 56.1 ± 17.7 | 3.2 ± 3.6 | 80 ± 46 | 2.3 ± 0.9 |
| SIB without KH | 251 | 236 | 5.6 ± 1.7 | 0.8 ± 0.7 | 1.3 ± 1.0 | 0.4 ± 0.5 | 58.3 ± 60.6 | 0.8 ± 2.7 | 107 ± 48 | 2.3 ± 1.3 |
| Erosion scarp | 36 | 18 | 4.9 ± 1.5 | - | 3.0 ± 5.7 | 0.08 ± 0.05 | 91.1 ± 29.5 | - | Various | - |
| Rill erosion trough | 31 | - | 4.4 ± 1.4 | - | 4.6 ± 5.8 | - | 118.6 ± 58.2 | - | Along the maximum slope | - |
| KH | 96 | 91 | 5.6 ± 1.3 | 2.0 ± 1.4 | 2.7 ± 2.3 | 0.29 ± 0.15 | 57.5 ± 23.3 | 4.8 ± 9.7 | 96 ± 51 | 2.2 ± 1.2 |
| KH with SIB | 48 | 17 | 5.4 ± 1.1 | 3.0 ± 1.4 | 4.3 ± 2.4 | 0.4 ± 0.2 | 57.6 ± 17.4 | 9.8 ± 10.9 | 89 ± 53 | 2.7 ± 1.0 |
| KH without SIB | 48 | 44 | 5.8 ± 1.5 | 1.6 ± 0.8 | 2.0 ± 1.1 | 0.2 ± 0.1 | 54.8 ± 26.6 | 2.6 ± 3.8 | 99 ± 47 | 1.9 ± 1.2 |

We did not find any particular statistical relations among the mapped elements and the topography. The qualitative granulometric analysis allowed to distinguish three grain size classes: (1) pebbles dominated (Figures 3a and 4); (2) sand dominated (Figures 3b and 4) and (3) composed by pebbles and boulders embedded in sandy matrix (Figures 3c and 4).

These three classes are not randomly distributed, but they are averagely sorted from class 1 to class 3 from the coastline to inland (Figure 4). Moreover, the elevation ranges of the grain sizes were different: class 1 ranged from 1.7 to 12.2 m, class 2 from 3.9 to 11.6 m, and class 3 from 2.5 to 10.5 m.

### 3.3. High Oceanic Wave Event Reconstruction

The high oceanic wave event that occurred during the austral summer 2018/2019, and was also responsible for the large number of SIBs identified during the early summer period of the 2019/2020 austral summer, has been reconstructed here with different data. The Landsat images suggested that the sea ice breakup at Edmonson Point occurred between the 3rd and 12th February and that the event occurred sometime between the 12th (date of the last Landsat image without ice blocks, not shown) and 17th February, because on the latter date the Landsat image shows how the whole beach was completely invaded by the SIBs. The largest ice blocks probably underwent little melting at the end of the summer season, therefore they remained visible until the following year (yellow arrow, Figure 5).

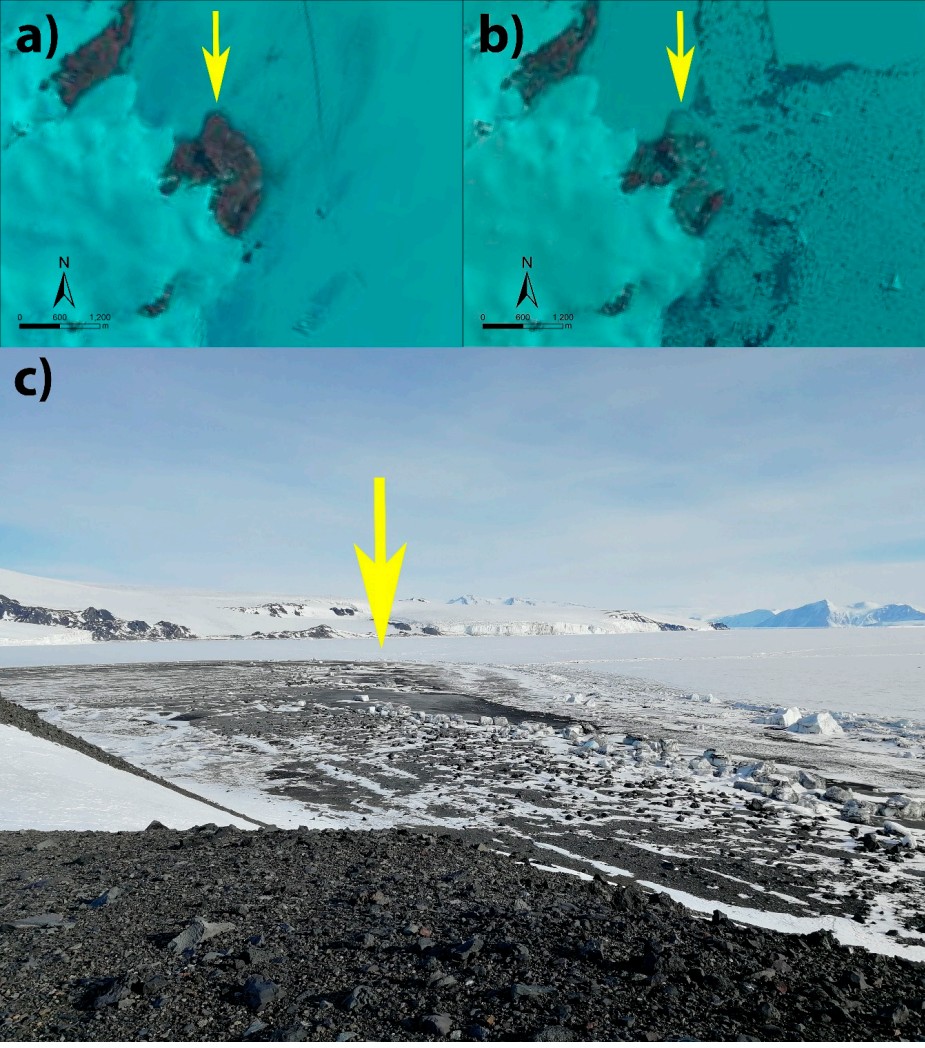

**Figure 5.** Temporal variation of the sea ice cover offshore and on the shore. Yellow arrows indicate the northernmost sector of the beach as reference in the Landsat images (**a**) 3 February 2019 and (**b**) 17 February 2019 and the ground view taken during the campaign (**c**) 11 November 2019.

In order to define the event at a more detailed temporal scale, by selecting the period 13–17 February 2019, it is possible to highlight the patterns of the soil surface temperatures (2 cm of depth) (Figure 6) at the different experiment sites (Figure 7). The pattern of the hourly temperatures shows how the zero-curtain period in which the temperature remained at 0 °C corresponded to the inundation time. Indeed, both the thermistors at OTC1c and TTv showed the beginning of inundation in the late morning of 15 February 2019, while the end of inundation in the late morning of 16 February 2019. The zero-curtain period clearly reflects the presence of liquid water that inhibits the temperature variations.

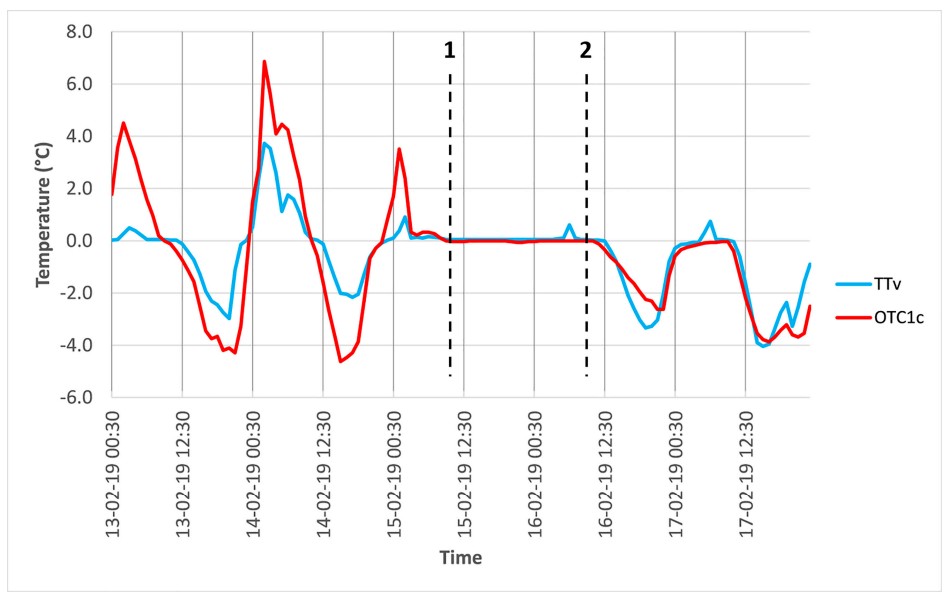

**Figure 6.** Ground surface temperature patterns between 13 and 18 February 2019 at the EPNB. The thermistors are all located at 2 cm of depth at different manipulation experiments (TTv = shallower side of a precipitation shield, OTC1c = exterior of an open-top chamber). 1 = beginning of the inundation, 2 = end of inundation according to the thermistors.

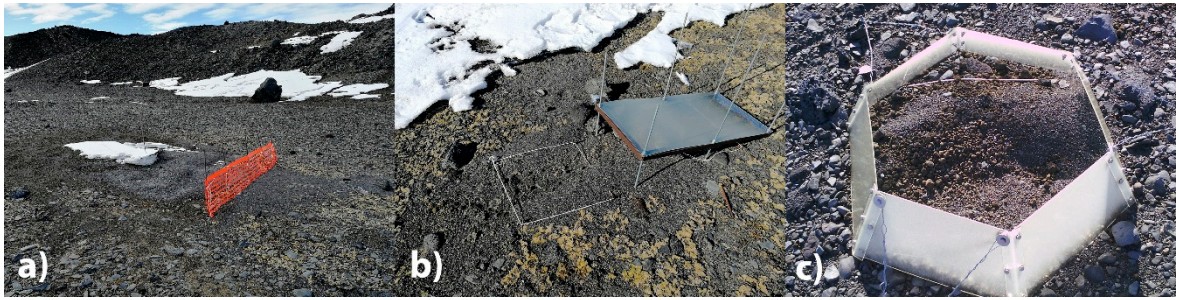

**Figure 7.** Details of the manipulation experiments that acted as obstacles and provided the accumulation of aeolian sediment. (**a**) SF, (**b**) TT, (**c**) OTC2. Please note that the mosses, especially within the OTC, are completely covered by gravel and sand.

Wind speed and direction were considered in this study, however the data were not useful for the aims of the paper. Indeed, wind speed and direction of the Maria AWS were only used to calculate the fetch just before the inundation time indicated by the soil thermistors (OTC and TT). At that moment (15 February 2019 00:00–02:00 UTC) the maximum wind speed coming from the open sea (E) reached 18 m s$^{-1}$. The fetch at that time corresponded to > 65 km with a sea surface composed of both open water and little fragments of sea ice. Therefore, the significant wave height resulted to be 1.95 m. According to online sources, the average sea level on 13 February 2019 00:00–12:00 UTC (moment A) was –1.716 m (respect to the Earth ellipsoid), while –1.722 on 15 February 2019 00:00–12:00 UTC (moment B) resulting in a decreasing sea water level of 6 mm from HYCOM. Hs

was averagely 1.18 m (2.2 m maximum) from the multi-grid data and averagely 1.08 m (1.27 m maximum) from the partition data of WAVEWATCH III® at moment A. At moment B, instead, Hs decreased: averagely 1.13 m (2.1 m maximum) from the multi-grid data and averagely 1.4 m (1.2 m maximum) from the partition data of WAVEWATCH III®. In addition, four elevation profiles were drawn at different locations from the coastline to the upper beach limit and crossing the beach ridges, in order to highlight the vertical pattern of different sectors of the beach (Figure 8). Overall, profile 4 was less elevated than the others and profile 2 resulted in the highest elevation. Considering the significant wave height (as indicator of the increased sea water level) of 1.95 m and that the coastline here is referred to as the pile-up sea ice zone (which is higher than the sea level), only the three northernmost transects showed a ZSS lower than 1.95 m (Figure 8a). Therefore, these sectors (from NW to NE) were overcome by the significant wave and thus indicate the possible way of the water entrance.

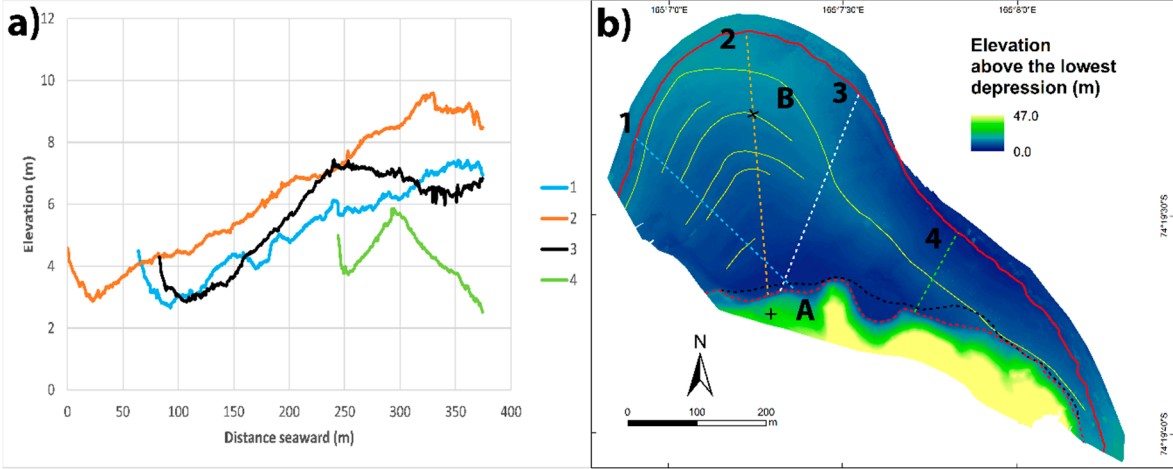

**Figure 8.** Four elevation profiles from the coastline to the upper beach limit crossing the beach ridges. (**a**) elevation profiles from the coastline to the upper beach limit, (**b**) location of the profiles on the DEM of the beach. "A" and "B" indicate the starting and ending point of the electrical resistivity tomography (ERT) showed in the Section 4.3. For other symbols refer to Figure 2. Please note that profile 1–3 are lower than the significant wave height (1.95 m) when considering the range between the coastline and the top of the main beach ridge.

### 3.4. Coast Dynamics

The cloud-to-cloud distance between the 1993 and 2019 surveys showed an interesting pattern of accumulation/erosion of the sediment. At a first glance, the spatial distribution of vertical changes followed the grain size throughout the beach. Indeed, the positive surface vertical changes (accumulation) distributed according to the grain size of the classes 1 and 2. The negative surface vertical changes (erosion), distributed shoreward, in correspondence of the class 3 (Figure 9).

The calculation of the annual rate of vertical change (cm yr$^{-1}$) by dividing the cloud-to-cloud distance for the span of time of the 2 surveys (26 years) showed that the range of annual rate varied between −0.0047 m yr$^{-1}$ (inland sector) and 0.063 m yr$^{-1}$ (coastline sector) for 95% of the pixels. The beach average resulted in 0.003 ± 0.032 m yr$^{-1}$, while the innermost sector (red) ranged between −0.047 and −0.03 m yr$^{-1}$, the central sector (orange) between −0.03 and −0.011 m yr$^{-1}$ and the shoreward sector (greens) between 0.011 and 0.063 m yr$^{-1}$.

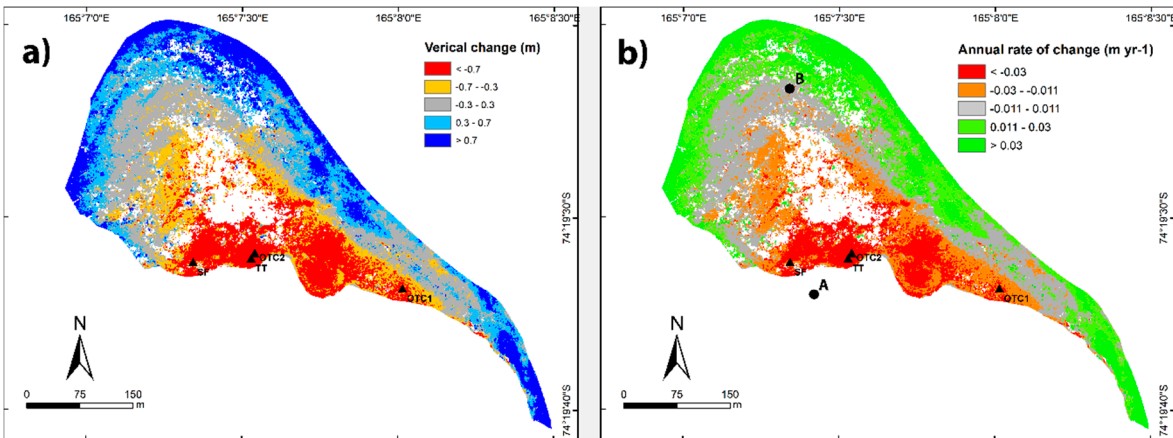

**Figure 9.** (**a**) Vertical change (m) occurred from 1993 to 2019, (**b**) average annual rate of vertical change (m yr$^{-1}$) from 1993 to 2019. White areas correspond to "no data" due to 3D reconstruction poor matching, while gray areas correspond to vertical changes or rate within the limit of detection of the clouds operation. "A" and "B" indicate the starting and ending point of the ERT showed in the Section 4.3.

Aeolian transport here can also be very efficient during the summer (and after snow melting) due to the several katabatic events that are not hindered by the uneven, rocky topography of the inland areas. The estimated annual accumulation of aeolian sediment was assessed using the manipulation experiments as sediment traps. In particular, the annual accumulation of the sandy to gravelly aeolian sediment ranged from 1.7 (TT) to 9.9 (SF) cm (see Table 3). The range of the average rate of change selected at the experiments location resulted by DEM comparison is between −3.6 and −5.4 (cm yr$^{-1}$) and its spatial pattern did not fit with the accumulation found between 2017–2018 in experiments. However, the data suggest that in all experiments there was an aeolian accumulation, possible only because the experiments were located in an area where there is wind erosion and consequently the traps decreased the wind rate causing the deposition of the sediment on their leeward side. The different shape of the traps probably conditioned the rate of accumulation that ranged between 1.7 and 9.9 cm yr$^{-1}$ (Table 3).

**Table 3.** Annual (2017–2018) averages (cm) of the aeolian sediment accumulation at the experiment locations on the EPNB and the mapped annual vertical rate (cm yr$^{-1}$) (Figure 9b) at the same locations. "n" refers to the number of replicates per site.

| Location | n | Mean (cm) | St.dev | Average Annual Rate (cm yr$^{-1}$) |
|---|---|---|---|---|
| TT | 3 | 1.7 | 0.1 | −5.4 |
| OTC2 | 3 | 3.1 | 1.4 | −4.5 |
| SF | 5 | 9.9 | 1.4 | −3.6 |

A detail of the aeolian accumulation found at the experiments is visible in Figure 7.

## 4. Discussion

### 4.1. Geomorphology and Morphometry

Among the aggradational features, the ice-push ridges are well documented in literature [8,19,68] and are clearly due to the pressure exerted by the sea-ice on the beach deposits along the coastline. Indeed, the highest concentration of this landform was found along the coastline, although some more inland features were mapped, probably as a consequence of very big and tabular SIBs floating and ploughing shoreward during the high oceanic wave event of 15 February 2019. The pebble ring structures are more peculiar, indeed they resembled the features described by Urdea [67] on the Svalbard Islands coasts but, contrarily to Urdea's findings [67] in which the ring was related to the melting of ground

ice, here these features were strictly associated to the SIBs action. In particular, floating SIBs with a keel that stranded on the substrate were able to plough the coarser sediment and accumulate it around the SIB. Moreover, the fact that all these features oriented from W to N indicates that they developed only during the backwash phase, when the keels of floating SIBs stranded on the beach surface and the remnant swash energy was only able to remove the finer material (Figure 10).

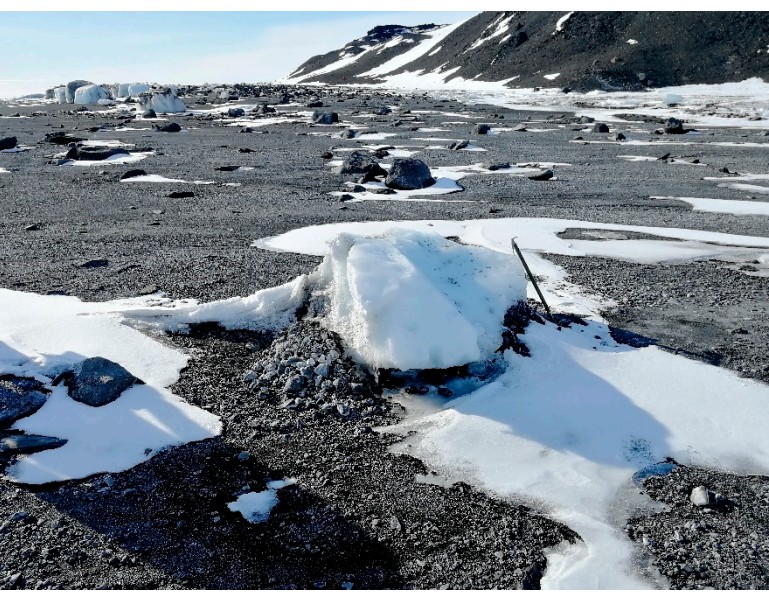

**Figure 10.** Example of coarser sediment (seaward) and finer sediment (inland) of a SIB that ploughed the substrate.

Pebble cones instead could be explained by a melting of smaller SIBs with gravel embedded in the ice. The steep slope of the coastline requires a high-energy geomorphic agent to develop, i.e., [30] and this reasonably evokes the action of the SIBs. Erosional forms differentiate between SIB-driven and water-driven. Unfortunately, drift-ice abrasion marks were not found on the beach, therefore the only erosional feature induced by SIBs were KHs. Indeed, the pressure of the SIBs on the substrate can produce depressions [27], the dimension of which did not relate with the dimensions of the correspondent SIB because their melting rate differed according to their albedo. As a result, KHs without SIBs were well distributed and underwent a faster melt out of the SIBs.

The water-flow-driven features are rill erosion troughs (and rill erosion areas) and erosion scarps. The former are landforms well known in other landscapes, e.g., [69], here developed by the sea water that probably flew according to the micro-topography (local slopes) and easily eroded the loose volcanic sediment. The latter were reasonably formed with minor energy because the pebbles (not iso-oriented) were laid on a finer matrix that was not washed out (Figure 3c). Further, water-driven erosional forms were distributed following the topography oriented towards the main stream of the lake.

### 4.2. High Oceanic Wave Event Reconstruction

From the satellite images and the thermistors data (TT, OTC1) the high oceanic wave event happened in the late morning of 15 February 2019. Sea water inundated the thermistors that showed a zero-curtain effect until the late morning of 16 February 2019. Since the soil temperature remained at 0 °C, it is more likely that a constant presence of liquid water was maintained in experiments until 16 February 2019, than the occurrence of multiple inundation events (wave run-up). This thermal state could be explained either: (a) by the wave run-up that lasted for one day and continuously supplied the beach with sea water because the volcanic sediments caused the water seepage, or (b) by the permafrost table

that was not sufficiently deep and allowed a saturation of the active layer. Indeed, the permafrost table depth during the days of the high oceanic wave event was ca. 20 cm in the closer experiment site located at less than 500 m of distance in the same lithology (data not shown). This is one of a few marine inundations where effects on the coast have been reported in Continental Antarctica, and was much larger than the previous shoreward limit of the sea surge indicated in this site by Simeoni et al. [27] that found a limit of 81 m from the coastline (no information about the water level increase and the wind speed were mentioned), while this event reached a maximum of 360 m shoreward.

The sea water level rise (here approximated with the Hs of 1.95 m) has been the agent that moved the SIBs shoreward and produced the drift-ice-driven landforms. Even though both the average sea level and Hs decreased during the study period (HYCOM and WAVEWATCH III®), maximum values of Hs resulted to be closer to the one obtained with Equation (1). This is probably due to the fact that the study area is a coastal area in a quite narrow bay (ca. 50 km wide), with respect to the dimension of the pixels (ca. $55 \times 15$ km) and therefore other inland factors might have affected our result. Alternatively, the formula used in this study did not account for climatic forcing but rather only fetch and wind. Moreover, the increase of the sea water level found from the calculated (and evident on shore) Hs and oppositely the decreasing trend of the SSH from moment A to moment B suggest that a storm surge did not occur during the high oceanic wave event. In support of this, no damages occurred at MZS due to waves or increased SSH during that period.

However, thermistor data precisely indicate the duration of the high oceanic wave event at a local scale. Similar SSH during storm surges are not uncommon in Greenland (1.25 m, [19]) and even higher in the Russian Arctic (3.87 m, [11]) but quite lower (0.20–0.41 m) in the Ross Sea [70]. Orford [7] suggested that wind speeds of 18 m s$^{-1}$ were not enough to generate overtopping swash flows on a beach crest in the UK of the same elevation of the one of EPNB (7.5 m). However, the effect of the ocean high-tide (up to 19.6 cm according to Byun and Hart [44]) during a storm surge could produce higher sea level rise [7] as probably occured during the studied high oceanic wave event. Indeed, in this sector of the Ross Sea other authors found a total inter-tidal variation of the sea level of 0.6 m at the Ross Island during storm surges [70].

The mapped landforms helped in defining the wave run-up shoreward limit. Indeed, all the mapped features indicated a flowing water process, so the wave run-up limit was set at the highest limit of the landforms' distribution. Moreover, the overwash (overpass of the main beach ridge) only occurred from NW to NE because the elevation difference between the coastline and the closest main ridge top was < 1.95 m. As a result, at the northernmost area of EPNB the number of mapped features was lower than the southernmost area. Therefore, the NW-NE sector can be defined as overwash area [16,24], while the SE sector as backwash area. Here, the water flow was not able to overpass the main beach ridge and concentrated the landforms at the seaward side of the main beach ridge. Further, the most frequent orientation of the drift-ice-driven features (KHs and SIBs) ranging between 80–104°, confirms that not only the sea water entered from NW–NE but also flooded the whole beach backwashing to E and reaching the lake at the shoreward side of the main beach ridge (Figure 8a, profile 4).

### 4.3. Coast Dynamics

In this paper we highlighted the exceptional impact of the high oceanic wave event of February 2019 on beach morphology. It is reasonable to think that similar events in the past could have modelled the beach topography, but in Antarctica, as well as in previously glaciated coasts, we have to take into account also the effect of the isostatic rebound after deglaciation in order to reconstruct the coast dynamics. Antarctic raised beaches are evident in this area [45] and on this coast [45,71] as beach ridges. However, from the coastline shoreward, we found decreasing reliefs/terraces (Figure 8a) that are in discordance to the classical raised beach profiles [35]. This opposite beach profile might be explained by the

only wave action that made the coastline prograde and generated a series of old berms depending on the intensity of the storm events [24].

The average surficial increase of $0.003 \pm 0.032$ m yr$^{-1}$ recorded in EPBN is slightly higher than the uplift rate found by Baroni and Orombelli [35] of 2–10 mm yr$^{-1}$ or Ivins and James [72] and Ivins et al. [73] of 0–2 mm yr$^{-1}$ in case of pure isostatic rebound. However, from the geomorphologic results, we suggest that the evolution of EPNB is likely due to the coupled effect of vertical uplift and high wave-energy events [30]. Indeed, a more intriguing possible explanation for the higher coastline sector surficial increase could be the subsurface sea water infiltrating and annually freezing at the permafrost table interface as aggradational ice that rises to the surface [74]. This subsurface sea water infiltration can effectively explain the low resistivity of the beach as shown in Figure 11. The electrical resistivity tomography (ERT) carried out in November 1996 next to the first 250 m of the profile 2 (Figure 8a,b) by one of the authors (MG) has been characterized by very high resistivity values (up to $10^6$ Ωm) in the innermost 60 m, between 220 m and the end of the profile and, subordinately, scattered along all the section but always above the current sea level. These values can be interpreted as a very cold and rich in ice permafrost with possibly some remnants of buried ice. Values below 10 kΩm above the sea level can be alternatively interpreted as open work gravel and pebble (as found in a shallow borehole carried out in 1996 that reached 2.5 m of depth approximately at the progressive 150 m) in which almost no ice was found but only loose material. Below the sea level between 60 and 220 m the values are lower than 1500 Ωm reaching values around 150 Ωm that can be interpreted in this kind of environment, e.g., [75] as: (a) saline frozen permafrost, or (b) infiltrated hypersaline brines, or (c) saturated gravel and sandy matrix, e.g., [76].

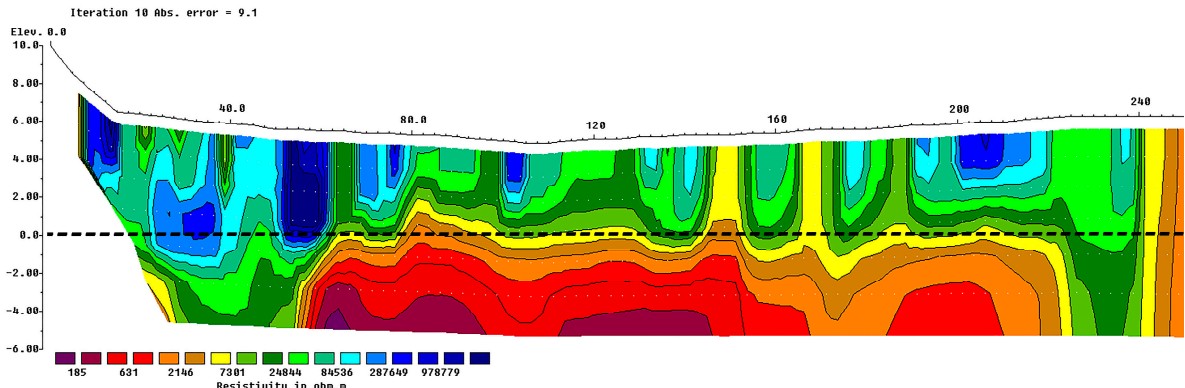

**Figure 11.** Electrical resistivity tomography carried out in Late November 1996 along the first 250 m of the profile 2 of Figure 8a,b oriented from south (**left**) to north (**right**). It is notable as a body of low resistivity (<700 Ωm) occurred in almost all the section under the sea level except for the innermost 60 m and the last 25 m. The low resistivity can be explained as frozen saline permafrost or unfrozen sediment saturated by hypersaline brines. Dashed line indicates the seal level.

Taking into account that the ERT was carried out in 1996 and that there are no clear evidences of thermokarst depressions, although a trend of active layer thickening [77–79] occurring in the last 20 years with a rate around 1 cm yr$^{-1}$ was recorded close to MZS, it is reasonable to think that: (a) a combination of permafrost degradation and wind erosion can be the driver of the subsidence in the innermost area where the ice content was higher in 1996 and the measured subsidence greater (between 0.08 and 0.03 m yr$^{-1}$), (b) probably wind action alone could explain the measured erosion rate in the central part of the beach where the ice content was lower and the subsidence more limited (0.03–0.011 m yr$^{-1}$). Further geophysical soundings and aeolian transport are required to confirm this hypothesis.

## 5. Conclusions

In this paper we demonstrated that remote sensing can be substantial in defining the dynamic of a high-latitude coastal environment where the role of cryogenic processes

like sea-ice [8,9] or permafrost [10] are the main drivers together with storm surge and wind action. Comparing the available satellite images for this beach—located at Edmonson Point (74° S) not far from the Italian Antarctic Station "Mario Zucchelli"—recorded in 1993 and our data obtained by helicopter survey carried out in summer 2019, we found an average surficial increase of $0.003 \pm 0.032$ m yr$^{-1}$ that is slightly higher than the uplift rate determined by Baroni and Orombelli [35] of 2–10 mm yr$^{-1}$ or Ivins and James [72] and Ivins et al. [73] of 0–2 mm yr$^{-1}$ in case of pure isostatic rebound. However, we suggest that the evolution of EPNB is likely due to the coupled effect of vertical uplift and high wave-energy events [30].

Indeed, the explanation for the seaward sector surficial increase could be related to the subsurface sea water infiltrating and annually freezing at the permafrost table interface as aggradational ice as suggested by the ERT carried out in 1996, suggesting the occurrence of saline frozen permafrost or hypersaline brines under the sea level and ice above the sea level.

Moreover, assuming that the trend of active layer thickening occurring in the last 20 years (ca. 1 cm yr$^{-1}$) not far from the study area is valid [77–79], the higher measured subsidence values (between 0.08 and 0.03 m yr$^{-1}$) in the innermost area can be explained by a combination of permafrost degradation and wind erosion while in the central part of the beach, where the subsidence is more limited (0.03–0.011 m yr$^{-1}$), wind action alone could explain the measured erosion rate.

In addition, regarding the current geomorphological processes, at EPNB the aggradational forms (112) were fewer than erosional forms (163) and SIBs resulted in being the most widespread features (315 blocks). It has been also noted that SIBs smaller than $0.28 \times 0.38 \times 0.11$ m were not able to produce any KHs.

Here, we also recorded in the late morning of 15-02-2019 the beginning of the inundation that lasted until the late morning of the next day. It was also possible to reconstruct that at that time the local water level rise (here approximated with the significant wave height) was 1.95 m.

To our knowledge this is one of the few marine inundations reported in Continental Antarctica which has had effects on the coast, and it is surely much larger (360 m from the coastline) than the previous limit of the sea surge indicated for EPNB by previous authors (81 m, [27]).

Despite it being common to see similar vertical limits of storm surge in the Russian Arctic (3.87 m, [11]), in the Ross Sea previous limits were much lower (0.20–0.41 m, [70]). Moreover, it is reasonable that to generate overtopping swash flows on such an elevated beach crest (7.5 m), the additional effect of the ocean tide should be evoked, especially considering that for this sector of the Ross Sea other authors found an inter-tidal sea level variation of 0.6 m at the Ross Island [70] during a storm surge.

**Author Contributions:** Conceptualization, investigation, data curation, formal analysis, validation were carried out by both the authors. Methodology, visualization and the original draft preparation were performed by S.P. The review and editing of the paper, the funding acquisition, project administration and its supervision was carried out by M.G. All authors have read and agreed to the published version of the manuscript.

**Funding:** Please add: This research was funded by PNRA (National Antarctic Program), grant number PNRA PNRA16_00194—A1 "Climate Change and Permafrost Ecosystems in Continental Antarctica" and PNRA18_00186-E "Interactions between permafrost and ecosystems in Continental Antarctica").

**Data Availability Statement:** The data presented in this study are available on request from the first author.

**Acknowledgments:** We want to thank all the logistical people of ENEA that support the research at Mario Zucchelli Station and Helicopter New Zealand and their pilots that allow this research. We also thank Emanuele Forte for a contribution in Figure 11 and Ulrich Neumann for his help in the field. We also are grateful to John Bills for his help in the manuscript phrasing.

**Conflicts of Interest:** The authors declare no conflict of interest.

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
