# Peer review of "Shore Evidences of a High Antarctic Ocean Wave Event: Geomorphology, Event Reconstruction and Coast Dynamics through a Remote Sensing Approach"

_remotesensing, doi:10.3390/rs13030518_

Round 1

Reviewer 1 Report

The manuscript has scientific interest, contains many new data obtained by various field and remote sensing methods. The study of the coastal dynamics in the polar regions under changing climate is undoubtedly of high relevance. After reading, there are a number of comments, questions and wishes, which are listed below, as well as in the text of the article.

I believe that new explanations, additions and answers to my questions will help improve the manuscript.

  1. Introduction

The introduction provides mostly general facts rather than specific data. What is known about the role of storm surges in the coastal dynamics in the polar regions nowadays? What is the height of the storm surges, at what strength of the wind and what are the consequences for the coastal dynamics.

I recommend to read one of the latest works, which considers an extreme storm surge in the Arctic:

Sinitsyn, A., Guegan, E., Shabanova, N., Kokin, O., Ogorodov, S. (2020) Fifty four years of coastal erosion and hydrometeorological parameters in the Varandey region, Barents Sea. Coastal Engineering, 157. 103610 [DOI] [WOS] [Scopus] [ResearchGate]

Lim M, Whalen D, J. Mann P, Fraser P, Berry HB, Irish C, Cockney K and Woodward J (2020) Effective Monitoring of Permafrost Coast Erosion: Wide-scale Storm Impacts on Outer Islands in the Mackenzie Delta Area. Front. Earth Sci. 8:561322. doi: 10.3389/feart.2020.561322

  1. Materials and methods

- Long-term regime of sea level (including tides) and wind (especially in ice-free period) according to data from nearby weather stations

The description of the work area lacks this information. Are sea level and wind observations carried out at the Mario Zucchelli station? From what year? Was this storm surge observed on it? What extreme levels and winds have been observed before?

- Methods

Some methodological points can be supplemented or transferred from the Results to this section (mapping, granulometric analysis and so on). It is very interesting to know, for example, how are the temperature loggers installed at a depth of 2 cm? Is there some kind of protection against possible erosion or ice impact if they are flooded during a storm surge? What is the installation elevation of the loggers?

- Elevation “zero”

It is not clear what is the “zero”, relative to which the elevations are given in the work.

Local elevation system? Mean sea level?

  1. Results and Discussion

- Figures

It is necessary to improve the readability of Figure 2 (many objects are hard to see).Here you can add profile lines (then you can avoid repeating in Figure 8) and the reconstructed surge boundary.

Figure 2 is sorely lacking in elevations.

Perhaps DEM is better as basemap than orthophoto image?

- Storm surge level

What is the formula for calculating significant wave height (1.9 m)?

Is this the height of the waves in the open sea, not in the coastal zone? It is not clear how this value relates to the storm surge level? I did not see a single section below significant wave height (1.9 m) on the profiles.

What was the storm surge level that flooded the loggers and brought SIBs? This level can be shown on profiles and on the map. From the position of the loggers on the map and the fact that they were flooded, I can assume that the storm surge level was close to the upper beach limit. Is it so? What is the elevation of the loggers? Have any consequences of flooding been identified at experiment sites (OTC1 and TT) or the strom surge passed without a trace and was recorded only in ground temperatures? Have there been any flooding effects on other experiment sites (OTC2 and SF).

Judging by Table 2, SIBs (very poorly seen in Figure 2) reached an elevation of 7.3 m, and ice-push ridges (not visible at all in Figure 2) - 8.9 m. So the storm surge level was more than 9 m? Is pebble cones elevation (up to 10.1 m; not visible at all in Figure 2) the evidence of the storm surge flooding? Then, the level was more than 10 m? Was the beach completely flooded, up to the upper beach limit?

- Ice-push ridges location

Table 2 shows an elevation of 6.9 ± 2.0 m. They are not visible in Figure 2. Figure 3a shows that they are at the coastline (near fast ice border), and it is indicated that they belong to the grain size zone of class 1 (judging by Figure 4, also along the coastline). How can there be such an elevation along the coastline? Or is there a mistake somewhere?

- Pebble cones location

Table 2 shows elevation 8.2 ± 1.9 m.They are not visible in Figure 2.Figure 3b indicates that they belong to the grain size zone of class 2 (judging by Figure 4, this is the middle part of the beach).How can there be such an elevation in the middle of the beach?Or is there a mistake somewhere?

- Landforms as evidences of storm surges

A short and clear explanation which Landforms and how help reconstruct storm surge (surge level, current direction, etc.) should be added.

- Grain size classes

Elevation ranges for each class can be added.

- Coastal dynamics

It is not entirely clear what the impact of the storm surge, and not by aeolian processes, permafrost degradation or isostatic rebound. What is the magnitude of the beach deformations as a result of the storm surge?

Did I understand correctly that all surface changes for the period 1993-2019 are explained by changes in permafrost: lowering of the surface - degradation of permafrost, and increase of the surface - aggradation of permafrost?

Can we say that the storm surge had only a local impact on the beach in the the individual landforms formation by sea ice or water (ice-push ridges, pebble cones, SIBs, KHs, erosion scarps, rill erosion troughs), and not an areal one?

11.12.2020                                                                                                                      

Author Response

Dear Reviewer, thank you for yoiur effort in improving our paper, this was very appreciated. We did almost all the corrections that you proposed and followed your suggestions. Below you can find the reply point to point of your suggestions and in addition in the manuscript in red you can find all the corrections and integrations.

The manuscript has scientific interest, contains many new data obtained by various field and remote sensing methods. The study of the coastal dynamics in the polar regions under changing climate is undoubtedly of high relevance. After reading, there are a number of comments, questions and wishes, which are listed below, as well as in the text of the article.

I believe that new explanations, additions and answers to my questions will help improve the manuscript.

  1. Introduction

The introduction provides mostly general facts rather than specific data. What is known about the role of storm surges in the coastal dynamics in the polar regions nowadays? What is the height of the storm surges, at what strength of the wind and what are the consequences for the coastal dynamics.

I recommend to read one of the latest works, which considers an extreme storm surge in the Russian Arctic:

Sinitsyn, A., Guegan, E., Shabanova, N., Kokin, O., Ogorodov, S. (2020) Fifty four years of coastal erosion and hydrometeorological parameters in the Varandey region, Barents Sea. Coastal Engineering, 157. 103610 [DOI] [WOS] [Scopus] [ResearchGate]

Thank you for your suggestions. We implemented the introduction and we include this important suggested reference. (i.e. new rows 43 and 45-50).

  1. Materials and methods

- Long-term regime of sea level (including tides) and wind (especially in ice-free period) according to data from nearby weather stations

The description of the work area lacks this information. Are sea level and wind observations carried out at the Mario Zucchelli station? From what year? Was this storm surge observed on it? What extreme levels and winds have been observed before?

Thank you for this comment. We added informations about the weather data availability in this area and the sea level observations.

- Methods

Some methodological points can be supplemented or transferred from the Results to this section (mapping, granulometric analysis and so on). It is very interesting to know, for example, how are the temperature loggers installed at a depth of 2 cm? Is there some kind of protection against possible erosion or ice impact if they are flooded during a storm surge? What is the installation elevation of the loggers?

Thank you for your suggestions. We added details about the loggers in that section and specified the grain size analysis. Anyway, the loggers are installed attaching to the downward side of a flat block of 2 cm of thickness and large diameter the thermistor. In this way the large but thin block prevent any  wind erosional effect and the large diameter (>20 cm usually)  of the block assure that the eventual frost heave did not affect the thermistor. (see also Guglielmin, 2006). However, we did not move the mapping results here because those are just a description of the findings and not a method description.

- Elevation “zero”

It is not clear what is the “zero”, relative to which the elevations are given in the work.Local elevation system? Mean sea level?

 We agree with you that this point is crucial and needed to be better explained. Since there was no liquid sea water but only grounded sea ice, during the helicopter fligths it was not possible to define a zero level as sea water surface. Therefore, we set the lowest depression (located north to the greater pond) as zero level and all the elevations cited are above that zero level. Moreover, setting zero level at the sea-ice – beach boundary would have conducted to elevation error because the sea-ice could have overpassed the seal level through the pile-up effect.

  1. Results and Discussion

- Figures

It is necessary to improve the readability of Figure 2 (many objects are hard to see).Here you can add profile lines (then you can avoid repeating in Figure 8) and the reconstructed surge boundary.

We agree with you. Indeed, we upload the figures at a better definition and we provided 2 sub-panels in figure 2 that enlarge the areas in which more of the smaller elements occurred allowing a better readability. We also added the reconstructed (via landforms limit) surge limit but we prefer to leave the profile lines in a separate figure together with the elevations pattern graph.

Figure 2 is sorely lacking in elevations.

Perhaps DEM is better as basemap than orthophoto image?

Thank you for this point. A background DEM would have been helpful but not as much as an orthophoto. Indeed, the colors of the basemap show the limit of the beach section such as the grain size sectors or the nearby texture of each landform. We rather added some elevation points. However, we added the DEM in Figure 8.

- Storm surge level

What is the formula for calculating significant wave height (1.9 m)?

Thank you. We added the formula.

Is this the height of the waves in the open sea, not in the coastal zone? It is not clear how this value relates to the storm surge level? I did not see a single section below significant wave height (1.9 m) on the profiles.

The formula used refers to restricted fetch applications without considering other variables like swell propagation and shallow-water effects. We confirm that profile 1 to 3 are below the significant wave height. Indeed, the profiles section between the sea-ice limit and the highest point are less than 1.95 m.

What was the storm surge level that flooded the loggers and brought SIBs? This level can be shown on profiles and on the map. From the position of the loggers on the map and the fact that they were flooded, I can assume that the storm surge level was close to the upper beach limit. Is it so? What is the elevation of the loggers? Have any consequences of flooding been identified at experiment sites (OTC1 and TT) or the strom surge passed without a trace and was recorded only in ground temperatures? Have there been any flooding effects on other experiment sites (OTC2 and SF).

Thank you. We added the surge limit in figure 2 and it is in fact near to the upper beach limit. We implemented the profiles and the map with the surge limit. Actually, the storm damaged the experiment sites: at SF the thermistor was ejected and the fence bent, ice blocks were found adjacent to TT and but no traces were found at OTC1. We added in the text (results) a sentence about the damaged experiments.

Judging by Table 2, SIBs (very poorly seen in Figure 2) reached an elevation of 7.3 m, and ice-push ridges (not visible at all in Figure 2) - 8.9 m. So the storm surge level was more than 9 m? Is pebble cones elevation (up to 10.1 m; not visible at all in Figure 2) the evidence of the storm surge flooding? Then, the level was more than 10 m? Was the beach completely flooded, up to the upper beach limit?

Thank you for pointing this out. All the landforms are located at an elevation above the lowest depression found on the DEM. Moreover, since the beach has a general immersion towards East, it is easy to find such elevation differences. According to us the beach was totally flooded almost up to the upper beach limit.

- Ice-push ridges location

Table 2 shows an elevation of 6.9 ± 2.0 m. They are not visible in Figure 2. Figure 3a shows that they are at the coastline (near fast ice border), and it is indicated that they belong to the grain size zone of class 1 (judging by Figure 4, also along the coastline). How can there be such an elevation along the coastline? Or is there a mistake somewhere?

We agree with you. We improved the figures. Such elevation at the “coastline” is possible because the limit of the sea-ice might not coincide with the sea level, therefore being more shoreward than the real coastline. We defined “coastline” better in Figure 2 caption.

- Pebble cones location

Table 2 shows elevation 8.2 ± 1.9 m.They are not visible in Figure 2.Figure 3b indicates that they belong to the grain size zone of class 2 (judging by Figure 4, this is the middle part of the beach).How can there be such an elevation in the middle of the beach?Or is there a mistake somewhere?

We provide here the same explanation of your previous comment.

- Landforms as evidences of storm surges

A short and clear explanation which Landforms and how help reconstruct storm surge (surge level, current direction, etc.) should be added.

Thank you. We implemented the discussion according this suggestion.

- Grain size classes

Elevation ranges for each class can be added.

Thank you, we added the ranges in the results.

- Coastal dynamics

It is not entirely clear what the impact of the storm surge, and not by aeolian processes, permafrost degradation or isostatic rebound. What is the magnitude of the beach deformations as a result of the storm surge?

We are sorry but from our result we can only infer what happened from 1993 to 2019. The only storm surge elevation effects are not detectable. It could be that during this span of time other destructive or constructive storms affected the beach but our findings focus on what we measured (i.e. permafrost, uplift, aeolian).

Did I understand correctly that all surface changes for the period 1993-2019 are explained by changes in permafrost: lowering of the surface - degradation of permafrost, and increase of the surface - aggradation of permafrost?

We cannot confirm your sentence. Indeed, the increase of elevation in seaward sector of the beach  could due to the combination of the permafrost aggradation and the accumulation of sediments transported by storms (and probably also the glacial isostatic rebound that still going on) whereas the subsidence of the landward sector is probably due to the wind erosion and permafrost degradation.

Can we say that the storm surge had only a local impact on the beach in the the individual landforms formation by sea ice or water (ice-push ridges, pebble cones, SIBs, KHs, erosion scarps, rill erosion troughs), and not an areal one?

We also cannot confirm this sentence. In fact, one evident areal impact is given by the rill erosion areas and, in addition, from our findings the storm flooded the whole beach but only in few points landforms developed due to differential water flow magnitude or substrate susceptibility. It might have been possible that a spatial distribution of vertical change occurred, but we do not have a DEM previous to the storm surge to verify it.

Reviewer 2 Report

Comments included in the PDF. In general, there are some wording changes that need to be made (signified as ‘Awk’ (for awkward statement).

I think the figures could be better. The ground photographs are striking of a beautiful field area, but the rest of the figures don’t do justice to the work, particularly the detailed mapping of the features. Try and connect the ground photos to the map locations as best you can

Remove the numerous excess line breaks, it makes the paper feel very choppy and hard to read.

Author Response

Dear Reviewer,

Thank you for your efforts in improving our paper. We followed your suggestions and made almost all the corrections that you requested.

We hope that now the paper can be acceptable. In the revised manuscript all the changes are in red while here the replies are in italic.

Comments included in the PDF. In general, there are some wording changes that need to be made (signified as ‘Awk’ (for awkward statement).

I think the figures could be better. The ground photographs are striking of a beautiful field area, but the rest of the figures don’t do justice to the work, particularly the detailed mapping of the features. Try and connect the ground photos to the map locations as best you can

Remove the numerous excess line breaks, it makes the paper feel very choppy and hard to read.

Thank you for your suggestions. We improved the images as you and other referees requested and we removed the line breaks. We would like to explain 3 points you underlined:

  • Your comment at old line 235 “Show graph”: we would prefer to remove the sentence and the equation since it is not fundamental for the aim of the paper and neither for the discussion.

  • We provided an adjustment for your comment at old line 314 “. state the values for aeolian deposition. Unless you are implying the gravel is aeolian in the next line???”. Indeed, at our sediment traps the aeolian sediment consisted of both sand and gravel. We can say that this gravelly sediments are aeolian in nature because we have also other years of data related to the sediment traps in which no storm occurred and therefore only the wind can blow in the open top chambers or accumulate on the leeward side of TT and SF also so coarse sediments. It is not really surprising here because the bigger clasts are of pumices (very light).

  • 7 shows the types of experiments (TT, SF and OTC) that act as barriers for the wind, therefore the wind became turbulent and decrease its speed allowing deposition of sediments within the OTC and on the leeward side of TT and SF. We think that the explanation gave in the text and the fig. 7 is enough considering that is not the main aim of this paper the aeolian transport measurement.

  • We would like to give an explanation to your comment at old line 425 “I think this is a bit of a reach given the data you have...”: This is a hypothesis that, according to us, fits the climatic circumstances of the study area. Indeed, both the processes could have occurred in the period and we can not quantify the role of each of them.

We did all the changes suggested

Reviewer 3 Report

This study evaluates the storm surge height, geomorphological changes and surface ground temperature from the composite analysis of satellite and UAV based imaginary, and temperature measurements. The methodology is clearly explained and results are sufficiently discussed. However, the validation for some reconstructed variables, such as storm surge, is not conducted. Therefore, the authors should validate their estimated storm surge height with other information in order to verify the accuracy of their method.

■ Major comments

  1. The authors should validate their reconstruction with other resources. The storm surge and wave height is now simulated globally. Wave height hindcast is conducted by Wave Watch 3 (https://polar.ncep.noaa.gov/waves/hindcasts/. Surge height is calculated by HYCOM (https://www.hycom.org/)
  2. The estimated values should be validated as much as possible. 

■ Minor Comments

Line 332. Unnecessary blank after "Indeed". This kind of blank is found in other places like line 438 and other places. 

Author Response

Dear Reviewer, 

Thank you for your usefull suggestions that were all assessed. We hope that you can find the paper now acceptable.

In the manuscript the changes are marked in red while below the replies are in italic.

This study evaluates the storm surge height, geomorphological changes and surface ground temperature from the composite analysis of satellite and UAV based imaginary, and temperature measurements. The methodology is clearly explained and results are sufficiently discussed. However, the validation for some reconstructed variables, such as storm surge, is not conducted. Therefore, the authors should validate their estimated storm surge height with other information in order to verify the accuracy of their method.

■ Major comments

  1. The authors should validate their reconstruction with other resources. The storm surge and wave height is now simulated globally. Wave height hindcast is conducted by Wave Watch 3 (https://polar.ncep.noaa.gov/waves/hindcasts/. Surge height is calculated by HYCOM (https://www.hycom.org/)
  2. The estimated values should be validated as much as possible. 

We agree with you. So we included both the validation methods and the compared results in the manuscript.

■ Minor Comments

Line 332. Unnecessary blank after "Indeed". This kind of blank is found in other places like line 438 and other places.

Thank you, we corrected the double spacings.

Reviewer 4 Report

Dear authors,

your paper represents one of the few examples of coastal geomorphology studies of high latitude beaches, I found it an interesting read.

However, I think that in the paper you use the wrong terminology for an increased total water level during a storm. In the paper you are confusing the effect of waves with the meteorological component. A surge is an increased water level above the predicted tide levels and consists of three contributions (the inverse barometer effect, the wind set-up, the wave set-up). You are confusing the wave contribution with the meteo contribution. Please clarify in the whole paper.

Author Response

Dear Reviewer, 

Thank you for your efforts in improving the manuscript we assessed all the points that you arised and we hope that the paper can be accepted in this form.

In the manuscript the changes are marked in red while below the replies are in italic.

Dear authors,

your paper represents one of the few examples of coastal geomorphology studies of high latitude beaches, I found it an interesting read.

However, I think that in the paper you use the wrong terminology for an increased total water level during a storm. In the paper you are confusing the effect of waves with the meteorological component. A surge is an increased water level above the predicted tide levels and consists of three contributions (the inverse barometer effect, the wind set-up, the wave set-up). You are confusing the wave contribution with the meteo contribution. Please clarify in the whole paper.

We thank you very much for your suggestion. We made all the manuscript consistently refer to significant wave height as one of the components of a storm surge( i.e. new rows 188-189.).

Round 2

Reviewer 3 Report

The reviewer satisfy the validation results between the author's measurement and online results provided by numerical modelling, especially estimated significant wave height which is well agreement with the observation. However, I have still one following question regarding the sea level of storm surge mentioned by the authors. 

Line 269-270: The authors mentioned that  "The average sea level on 13/02/19 00:00-12:00 UTC (moment A) 268 was -1.716 m (compared to the global mean sea level), while -1.722 on 15/02/19 00:00-12:00 UTC 269 (moment B) resulting in a decreasing sea water level of 6 mm from HYCOM".

The HYCOM numerical modelling suggests that the sea level is negative compared with ground level (0 m). However, when storm surge happens, sea level must be positive value (like over 1.0 m) from the mean sea level considering the definition of storm surge (see https://www.nhc.noaa.gov/surge/).

Therefore, "storm surge" mentioned by authors might be "high oceanic wave event" without an increasing in sea level under storm. The land inundation might be caused by the associated wave run-up. If so, please consider to use the word of "high oceanic wave event" instead of "storm surge" in the title as well as in many other contexts. And please to consider to include the word of "wave run-up" in the context. 

Author Response

Dear Reviewer

Thanks for this further request, our reply is below in italic while changes in the manuscript are marked in red.

The reviewer satisfy the validation results between the author's measurement and online results provided by numerical modelling, especially estimated significant wave height which is well agreement with the observation. However, I have still one following question regarding the sea level of storm surge mentioned by the authors. 

Line 269-270: The authors mentioned that  "The average sea level on 13/02/19 00:00-12:00 UTC (moment A) 268 was -1.716 m (compared to the global mean sea level), while -1.722 on 15/02/19 00:00-12:00 UTC 269 (moment B) resulting in a decreasing sea water level of 6 mm from HYCOM".

The HYCOM numerical modelling suggests that the sea level is negative compared with ground level (0 m). However, when storm surge happens, sea level must be positive value (like over 1.0 m) from the mean sea level considering the definition of storm surge (see https://www.nhc.noaa.gov/surge/).

Therefore, "storm surge" mentioned by authors might be "high oceanic wave event" without an increasing in sea level under storm. The land inundation might be caused by the associated wave run-up. If so, please consider to use the word of "high oceanic wave event" instead of "storm surge" in the title as well as in many other contexts. And please to consider to include the word of "wave run-up" in the context. 

Although to define the cause of the wave inundation of EPNB it was not our main aim we agreed with You that with the available data it is not correct to define this event as a Storm Surge and therefore we changed the manuscript accordingly.